# Epidemiological and Public Health Significance of *Toxoplasma gondii* Infection in Wild Rabbits and Hares: 2010–2020

**DOI:** 10.3390/microorganisms9030597

**Published:** 2021-03-14

**Authors:** Sonia Almeria, Fernando H. A. Murata, Camila K. Cerqueira-Cézar, Oliver C. H. Kwok, Alicia Shipley, Jitender P. Dubey

**Affiliations:** 1Office of Applied Research and Safety Assessment (OARSA), Center for Food Safety and Applied Nutrition (CFSAN), Food and Drug Administration, 8301 Muirkirk Road, Laurel, MD 20708, USA; 2Animal Parasitic Disease Laboratory, Agricultural Research Service, United States Department of Agriculture, Beltsville, MD 20705-2350, USA; fernandomurata@hotmail.com (F.H.A.M.); camilakcerqueira@gmail.com (C.K.C.-C.); Oliver.Kwok@usda.gov (O.C.H.K.); Jitender.Dubey@usda.gov (J.P.D.); 3Joint Institute for Food Safety and Applied Nutrition, University of Maryland, College Park, MD 20740, USA; Alicia.Shipley@fda.hhs.gov

**Keywords:** *Toxoplasma gondii*, hares, rabbits, last decade, review

## Abstract

Toxoplasmosis is a zoonosis of global distribution, and *Toxoplasma gondii* infections are common in humans and animals worldwide. Hares and rabbits are important small game species, and their meat is consumed by humans in many countries. Demand for rabbit meat for human consumption is increasing; therefore, toxoplasmosis in rabbits and hares is of epidemiological significance. Viable *T. gondii* has been isolated from rabbits. The present review summarizes worldwide information on the seroprevalence, parasitological investigations, clinical cases, isolation, and genetic diversity of *T. gondii* in wild rabbits, free domestic rabbits, hares, and other rabbits from 2010 to 2020. Differences in prevalence, susceptibility, genetic variants, and clinical implications of *T. gondii* infection in rabbits and hares are discussed. This review will be of interest to biologists, parasitologists, veterinarians, and public health workers. Additional studies are needed to increase our knowledge of genetic variants and the population structure of *T. gondii* in rabbits and hares and to understand the differences in susceptibility to *T. gondii* in hares in different areas.

## 1. Introduction

Toxoplasmosis is a zoonosis of global distribution caused by the protozoan parasite *Toxoplasma gondii,* the only species in the *Toxoplasma* genus. *Toxoplasma* can infect virtually every warm-blooded animal, including humans and livestock, as intermediate hosts [1]. Felids (domestic and wild) are the definitive hosts, and the only hosts able to excrete oocysts to the environment. *Toxoplasma gondii* is considered one of the most important foodborne and waterborne parasites of veterinary and medical importance worldwide [2,3].

Consumption of undercooked or raw meat containing tissue cysts is the primary way of infection of *T. gondii* for humans [1,4]. Humans can also become infected by consuming food or drinking water contaminated with oocysts or by accidentally ingesting oocysts from the environment. In addition, humans, especially hunters, may also acquire infection through contact with the parasite while field dressing game [1,5,6].

After being consumed in uncooked or raw meat containing tissue cysts (carnivores) or in feed or drink contaminated with oocysts (all warm-blooded animals), *T. gondii* initiates intestinal replication. Bradyzoites and sporozoites, respectively, are released and infect the intestinal epithelium [1], and then tachyzoites are disseminated to other tissues via the bloodstream and lymph throughout the body and replicate intracellularly until the cells break and cause tissue necrosis. Young and immunocompromised humans and animals may succumb to generalized toxoplasmosis at this stage. Older humans and animals mount a powerful, cell-mediated immune response to the tachyzoites (mediated by cytokines) and control infection, driving the tachyzoites into bradyzoites inside tissue cysts. Tissue cysts are usually seen in neurons but also in the liver, heart, brain, tongue, diaphragm, and skeletal muscle [1]. Immunity does not completely eradicate infection, and *T. gondii* can remain viable in tissues for many years, possibly for the life of the host [1]. Although there is a robust innate immune response elicited during infection, *T. gondii* has evolved strategies to successfully overcome or manipulate the immune system, with mechanisms that include control of host gene transcription and dysregulation of signaling pathways, modulation of cell adhesion and migration, secretion of immunoregulatory cytokines, and apoptosis. Many of these host–pathogen interactions are overseen by parasite effector proteins secreted from the apical secretory organelles, including the rhoptries and dense granules [7].

Hares (genus *Lepus*) and rabbits (mainly the European rabbit (*Oryctolagus cuniculus*)) are important small game animals extensively hunted in many countries. Rabbit meat, considered one of the most nutritional white meats, is very popular in many countries [8] where the demand for human consumption of rabbit meat is increasing. In fact, breeding of domestic rabbits for human consumption has become traditional in many countries in Europe, South America, and Asia [5,9,10,11]. Rabbits are broadly distributed and generally abundant. They are indigenous and inhabit a wide range of environments in all continents except Antarctica and Australia. In Australia, the European rabbit was introduced in 1859 for hunting, and it is now considered a pest [12]. Some other rabbit species, such as the Eastern cottontail rabbit (*Sylvilagus floridanus*), were introduced from North America to some parts of Europe to increase the potential for small game hunting but are now considered an invasive species [13]. When referring to hares, the European brown hare (*Lepus europaeus*) is probably the most important game animal in Europe [6]. In China, another type of hare, the Tolai hare (*Lepus tolai*), is very popular due to its rich nutritional value [14].

The isolation of *T. gondii* from rabbits was of historical significance. A *T. gondii*-like parasite was first described in 1908 by Splendore in a rabbit (*O. cuniculus*) in Brazil at the beginning of the twentieth century and the same year in a rodent, *Ctenodactylus gundi*, from Tunisia by Nicolle and Manceaux [1]. This parasite was originally considered *Leishmania* sp. and was subsequently named *Toxoplasma gondii* in 1909 by Nicolle and Manceaux [1]. The rabbits were captive in the laboratory of Splendore, and no clinical signs were observed before death [1]. In addition, rabbits have also been used to characterize different *T. gondii* antibodies and to standardize serological tests [15,16,17].

Rabbits and hares are herbivores and are mainly infected with *T. gondii* via the ingestion of water and plants (grasses, weeds, and crops) containing oocysts excreted by domestic cats or free-ranging felids with which they share the same habitats [1,18,19]. This makes rabbits useful as sentinel species for the level of environmental contamination with *T. gondii* oocysts [17,20].

Infected rabbits and hares can act as a potential source of *T. gondii* for other animals, especially for carnivores, including cats, but also for humans [1,11,21]. Humans may become infected by eating undercooked or raw rabbit meat or from hand-to-mouth after processes such as slaughtering and skinning rabbits [22]. There are few regulations for commercial rabbit meat and a lack of legal control in many countries regarding the slaughter of rabbits for human consumption [23]. Therefore, toxoplasmosis in rabbits and hares is of public health and epidemiological significance.

This review provides up-to-date information on the seroprevalence, clinical cases, diagnosis, and genotypes present in wild rabbits, free domestic rabbits, and hares in the last decade (2010–2020). Studies on *T. gondii* infection in domestic rabbit from breeding farms and slaughtered animals for human consumption have been included for comparison and to give a complete view of the public health significance of *T. gondii* infection in these species. The source of rabbit meat (hunted or domestically raised) is often not specified in grocery stores or on a restaurant menu. Therefore, we have reviewed information on both domestic and hunted rabbits. The literature cited is from 2010–2020; information before 2009 was reviewed previously [1,24].

## 2. Seroprevalence of *T. gondii* in Rabbits and Hares

### 2.1. Serological Investigation in Rabbits

Detection of antibodies is important for epizootiological studies in rabbits because most infections are subclinical [1]. Serum antibodies to *T. gondii* have been found in rabbits, mainly in the European rabbit (*O. cuniculus*)*,* in several surveys worldwide. Data from 2010 to 2020 are summarized in Table 1. Antibodies to *T. gondii* in rabbits were reported ranging from 0.9% to 37.5%, with most studies showing seroprevalence levels between 10–15% (Table 1).

The highest prevalence in domestic rabbits (*O. cuniculus*) was reported in Egypt: 26.7% [29] and 37.5% [30], respectively. IgM antibodies were observed in nine slaughtered rabbits (6.0%) as an indication of active infection [30]. Additionally, in Egypt, antibodies against *T. gondii* were detected in 11.3% of farmed rabbits [31] (Table 1). High seroprevalence (23.4%) was also observed in free domestic rabbits in a rural area in China, and viable *T. gondii* was isolated [28] (Table 1).

In free wild rabbits, the most comprehensive survey was from Australia and involved 2114 wild rabbits (*O. cuniculus*) sampled over a 12-year span [20]. Mean seroprevalence was 9.9%. Lower seroprevalences were found in wild rabbit studies in Europe: 3.3% of 548 wild rabbits in Scotland [36] and 2.8% of 36 wild rabbits in Portugal [35] (Table 1). Similarly, in another rabbit species, the wild cottontail rabbit (*S. floridanus*), low seroprevalence of *T. gondii* antibodies (2.1%) was observed in Italy [13]. Wild rabbits and domestic rabbits that have access to the outdoors are likely more susceptible to *T. gondii* infections than pet rabbits confined indoors due to the possibility of direct contact with the contaminated environment [17].

Animal management (living and rearing outdoors or in a pet shop), the possibility of consuming contaminated vegetables, and cohabitation with cats are the main risk factors that could account for some of these observed differences. Additionally, different serological tests used, types of samples analyzed, cut-off titers, and sample sizes could contribute to these differences as well.

The Indirect fluorescent antibody test (IFAT) is a well-established technique for detecting anti-*T. gondii* antibodies in different animal species, including rabbits, but requires conjugate and is not automated. ELISA has the advantage of being automated and allows the testing of many samples, but also needs species-specific conjugates. A commercial ELISA is available for rabbits (see legend Table 1). Therefore, many studies in rabbits and hares rely on agglutination tests that do not need species-specific secondary antibodies. The specificity, sensitivity, and cut-off value of these serological tests have been validated in a few studies (see Section 5). Development and standardization of diagnostic tests for *T. gondii* infection are still necessary [3].

### 2.2. Seroprevalence in Hares

In the last decade, there have been several reports of *T. gondii* antibodies in hares from Europe as well as in two studies from China (Table 2).

The European brown hare (*L. europaeus*) was the most studied hare species. Prevalence of anti-*T. gondii* antibodies ranged from 0% to 21% among studies in different countries (Table 2). The low seroprevalence (0% in *L. europaeus* and 4.2% in *L. timidus*) observed in Finland, concomitantly with cases of fatal toxoplasmosis, might indicate a natural, inherent susceptibility of these hare species to *T. gondii* at least in those areas [40]. On the other hand, higher seroprevalence levels have been observed in other countries, e.g., France, Austria, Czech Republic or Slovakia [21,41], suggesting that European brown hares can survive *T. gondii* infection in the wild in other areas [6]. In another hare species, the Iberian hare (*Lepus granatensis*) endemic in Spain, no clinical cases were detected, and a moderate to low seroprevalence (11.4%) was observed [5]. In that study, significantly higher seroprevalence was observed in juvenile Iberian hares compared with the adult ones [5], and the authors suggested that possibly *T. gondii* infection affected the survival of infected hares, and thus infected juveniles reached the adult stage at lower rates than the non-infected hares or that a short-lived humoral immune response against *T. gondii* existed in this species [5].

## 3. Isolation of Viable *T. gondii* DNA from Rabbits and Hares

Viable *T. gondii* was isolated by bioassay from rabbits in Argentina [44,45], Brazil [23], and China [28] in the last decade (Table 3). In Argentina, isolation was performed in an aborted rabbit dam [44,45] with disseminated toxoplasmosis (Table 3). Two seropositive rabbits in Brazil were bioassayed in mice and in cats, and one cat excreted *T. gondii* oocysts after consumption of one of the rabbit’s brain. Both strains were pathogenic to mice [23]. In China, viable *T. gondii* was isolated from one seropositive rabbit. The isolate was also pathogenic to mice [28]. Little is known about the genetic diversity of *T. gondii* in rabbits and hares, but three of the four isolates characterized by PCR-RFLP markers were different genotypes (Table 3).

## 4. Detection of *T. gondii* DNA in Tissues of Rabbits and Hares

The most frequently used target genes for *T. gondii* PCR in both humans and animals are the repetitive regions of the 35-copy B1 gene and 300-copy 529 bp repetitive element. *Toxoplasma gondii* DNA was detected in three (2.4%) of 126 domestic rabbits (*O. cuniculus*) from Zhengzhou, eight of 248 (3.2%) from Luoyang, and two of 96 (2.1%) rabbit brains and hearts from slaughtered rabbits from food markets in China [7] (Table 4). A very high prevalence of infection was recently reported in European wild rabbits from central Portugal, with 67.9% of 28 hunted rabbits showing *T. gondii* DNA based on a nested PCR targeting the surface antigen 2 (*SAG2*) gene [46]. *T. gondii* DNA was also detected in tissues of three (brain of one, skeletal muscle of two) of 144 wild cotton-tailed rabbits (*S. floridanus*) [13]. The tissues were collected from animals culled for demographic control of this species in Italy [13]. Shin et al. 2013 [32] found 16.2% of 142 rabbits to be positive for the presence of *T. gondii* DNA at higher levels than those shown by serological levels by enzyme-linked immunosorbent assay (ELISA) (10.6%). In this study, differences were observed among breeds with higher prevalence in Flemish giant and chinchilla rabbits compared to crossbreeds and New Zealand white rabbits [31]. On the contrary, a nested PCR of blood from pet rabbits in Poland showed all tested blood samples (n = 360) were negative for the *T. gondii B1* gene in these pet rabbits [33].

*Toxoplasma gondii* DNA has also been detected in hares (Table 4). Jokelainen et al. (2011) [40] detected *T. gondii* DNA in hares that died of visceral toxoplasmosis. Račka et al. (2020) [19] detected *T. gondii* using a qPCR targeting the 529 bp repetitive region in the tissues of two of 36 hares killed or found dead (5.5%). One of the infected hares had high levels of parasites in the spleen and lung, a unique finding. *Toxoplasma gondii* DNA was also detected in the Tolai hare (*Lepus tolai)* in three different areas of China [13] (Table 4). On the other hand, *T. gondii* infection was not found by PCR in the liver of seropositive Iberian hares (*L. granatensis*) in Spain [5] or in livers of *L. europaeus* in Greece [40]. In the latter species, Aubert et al. (2010) [41] were not able to detect viable *T. gondii* from heart tissues using a bioassay in mice; it should be noted that only three seropositive *L. europaeus* were bioassayed.

## 5. Validation and Comparison of Serological Results by Different Techniques in Rabbits

Little is known of the validity of different serological tests for the detection of *T. gondii* antibodies in rabbits. In a recent study, validation of the modified agglutination test (MAT) for detection of *T. gondii-* specific antibodies was performed using wild rabbit samples and was compared to an indirect fluorescent antibody test (IFAT) as a reference test [20]. The two assays showed 94.3% agreement, with the MAT performing with a sensitivity of 84.1% (95% CI: 69.9–93.4%) and specificity of 96.7% (95% CI: 93.0–98.8%) relative to the IFAT.

In a study on pet rabbits in Japan, 337 serum samples were tested by different serological methods; initially, IgG and IgM ELISAs were performed, and those samples seropositive by ELISA were further analyzed using a latex agglutination test (LAT), Western blotting (WB), and an indirect immunofluorescence assay (IFAT) [17]. The rates of seropositivity for *T. gondii* were 0.89% (3/337) and 0.29% (1/337) for the IgG and IgM ELISAs, respectively. SAG1 and SAG2 were detected as major antigens by the positive rabbit sera in WB analyses and were associated with strong staining observed by IFAT in *T. gondii* tachyzoites [17]. In pet rabbits from breeding farms in Korea [32], whole blood was analyzed by serology, nested PCR, and immunoblotting (three samples negative and six samples positive based on ELISA). Interestingly, there was a higher *T. gondii* prevalence as detected using nested PCR (16.2%) relative to ELISA (10.6%). The authors suggested that the nested PCR-positive/ELISA-negative rabbits might have harbored prepatent or latent infections with *T. gondii.* All PCR-positive samples were sequenced and corresponded (99% homology) to partial *T. gondii* B1 gene sequences.

## 6. Clinical Infections

### 6.1. Cinical Cases in Hares and Rabbits in the Last Decade

Infections in rabbits are documented to be mainly subclinical [36]. On the other hand, since the 1950s, high susceptibility and fatal systemic toxoplasmosis cases have been reported in European Brown hare (*L. europaeus*), mainly from Scandinavia, where harsh winter conditions are common [1,6,40]. To our knowledge, these epizootics have not been seen in other countries. Reports of clinical toxoplasmosis in rabbits and hares, preceding the year 2010, were reviewed earlier [1]. Only two studies reported acute toxoplasmosis in a few rabbits in the USA [1]. Since then, a retrospective report of fatal acute disease caused by *T. gondii* was diagnosed in a domestic rabbit in Brazil in which *T. gondii* was identified by immunohistochemistry (IHC) in histological sections of spleen and liver tissue and genotyped [11] (Table 5). The animal was from a farm where there were 400 young rabbits (approximately 40 to 45 days old) that developed diarrhea and severe dehydration after weaning. A total of 150 animals died during a 2-month period, 24 h after the onset of clinical signs [11].

As indicated above, an outbreak of abortion occurred on a small rabbit farm in Argentina [44]. A rabbit that had aborted was necropsied; there was no information on the aborted fetus. The rabbit had disseminated toxoplasmosis, and the diagnosis was confirmed by IHC testing, PCR, and isolation of viable *T. gondii* [45] (Table 3).

In hares, in a retrospective study of *L. europaeus* and *L. timidus* that died of visceral toxoplasmosis, the results were confirmed by positive serology and IHC. The proportional mortality rates and the *T. gondii* antibody prevalence differed significantly between the two host species. Limited genotyping revealed ToxoDB genotype #1 (type II) in those animals [40]. In another study, two hares were found dead in the Czech Republic and macroscopic observations indicated acute toxoplasmosis [19]. One hare had a very high number of *T. gondii* parasites detected in the lungs and spleen, exceeding 1,000,000 parasites/gram. Both studies on *L. europaeus* hares, in Finland and in the Czech Republic, revealed the genotype to be *T. gondii* type II clonal lineage [19,40].

### 6.2. Histopathology and Immunohistochemistry

Microscopic lesions of *T. gondii* infection in rabbit tissues (brain, heart, and diaphragm) revealed mainly granuloma, mononuclear cell infiltrates, and degeneration areas; necrosis was observed in brain and heart tissues [26].

Splenomegaly seems to be a common finding for rabbits and hares with fatal toxoplasmosis [11,40]. In a retrospective study of spleen and liver tissue of a domestic rabbit with fatal acute disease caused by *T. gondii* in Brazil, the spleen was grossly enlarged and had extensive necrosis of parenchyma [11], as well as a marked quantity of transparent fluid in the abdominal and thoracic cavity, likely attributable to hepatomegaly and congestion. In hares with very high parasitic burdens [19], the most typical gross findings in *T. gondii* positive animals were pulmonary congestion, splenomegaly, and granulomatous and necrotic lesions in the liver. As indicated above, in this study, a high number of *T. gondii* parasites were detected, especially in the lungs and spleen (more than 1,000,000 parasite counts per gram of tissue). In other hare studies, the pathomorphological findings consisted of enormous enlargement of the spleen, lung edema, enlarged mesenterial lymph nodes and paleness, an uneven color, and/or hemorrhagic areas in the liver [40]. Most of the animals had an otherwise good body condition.

The *Toxoplasma gondii* load in tissues of asymptomatic animals is low [1], thus the chances of detection of the parasite by histopathology and IHC are also low. To our knowledge, only a few studies [11,40,45] have evidenced the presence of *T. gondii* in rabbits and/or hares by IHC in the last decade. Jokelainen et al. (2011) [40] performed retrospective IHC on formalin-fixed liver tissue samples from hares with fatal toxoplasmosis as the recorded cause of death, and toxoplasmosis was confirmed by IHC and positive serology in 8.1% (14 of 173) of brown hares (*L. europaeus)* and 2.7% (4 of 148) of mountain hares (*L. timidus*) tested. The liver was chosen because it is typically affected in fatal toxoplasmosis in hares, and it has successfully been used as the organ of choice for IHC. Based on IHC detection of the parasite in liver tissue, 17 of 18 (94%) cases originally diagnosed as fatal toxoplasmosis of hares were retrospectively confirmed, and one new case was found. Only one hare with recorded toxoplasmosis was IHC negative.

In the rabbit that had aborted at a farm in Argentina, disseminated toxoplasmosis was confirmed by IHC testing, PCR, and isolation of viable *T. gondii* [45] (Table 3). *T. gondii* was also demonstrated retrospectively by IHC in histological sections of spleen and liver tissues of one rabbit suffering fatal acute toxoplasmosis in Brazil [11].

## 7. Genotypes of *T. gondii* in Rabbits and Hares

### 7.1. Genotypes Based on Isolation of Viable T. gondii

In the studies that isolated viable *T. gondii* from domestic rabbits by bioassays (Table 3), the genotypes reported were #48 from Argentina, #11 and #19 from Brazil, and #2 from China [23,27,44].

### 7.2. Genotypes Based on Isolation of T. gondii DNA from Tissues

Based on genotyping in tissues in the last decade, different genotypes were observed in rabbits and hares. In rabbits, interestingly, in a household where raw rabbit meat was eaten and cases of human toxoplasmosis occurred, a virulent strain of the parasite was isolated from the brain of a rabbit fed green fodder from the home garden in Poland, but no details were given by Sroka and Szmańska (2012) [34].

In the case of fatal acute toxoplasmosis in a domestic rabbit in Brazil [11], genotyping of DNA isolated from the frozen spleen revealed nonarchetypal ToxoDB genotype #8 [11], type BrIII genotype, which is a typical clonal Brazilian lineage (Table 4); this was the first description of acute disease associated with this genotype in a non-human naturally infected animal host. The genotype found in Brazil reported the first occurrence of the 291 allele for the typing marker TUB2 in a type BrIII strain, emphasizing that isolates homogenously classified as type BrIII via PCR-RFLP can be genetically diverse as well as the genetic diversity of *T. gondii* in Brazil [11].

As indicated above, in the European brown hares, ToxoDB genotype #1 (type II clonal lineage) has been reported from clinical samples in Finland [40] and the Czech Republic [19]. Both studies used microsatellites for genotyping; six microsatellites were used in the Finnish study, and 15 microsatellites were used in the Czech report. In the Czech Republic, one *T. gondii* isolate was archetypal clonal type II and the other a type II variant (W35 = 244). The animal infected with Type II variant W35 had severe disseminated toxoplasmosis, with an estimated 7 million *T. gondii* per gram of tissue [19].

In China, isolation and genotyping of *T. gondii* DNA in two studies, one in rabbits and the second in Tolai hares, found genotype #9 [7,14]; this genotype has been previously identified in several hosts, including domestic animals in China [14].

## 8. Conclusions

Although in the last decade there have been several studies performed on the epidemiology and clinical features of *T. gondii* in domestic and wild rabbits and hares, more studies on the prevalence and clinical and immunological implications of *T. gondii* infection in rabbits and hares are needed. Additional studies would also help to understand the differences in susceptibility to *T. gondii* in hares in different areas and to increase our knowledge of genetic variants and the population structure of *T. gondii* in rabbits and hares. More studies of the specific factors that exacerbate disease in rabbits and hares are also needed.

Viable *T. gondii* could be present in the organs of seropositive rabbits and hares. Therefore, hunters should be aware of *T. gondii* infection in clinically healthy rabbits and hares and are advised to take precautions while field dressing game and to freeze or cook rabbit and/or hare meat before human consumption [5]. In domestic rabbit farms, it is important to implement good hygienic conditions and, if possible, prevent the presence of cats and dogs on the farms [18].

## Figures and Tables

**Table 1 microorganisms-09-00597-t001:** Seroprevalence of *Toxoplasma gondii* in rabbits from 2010 to 2020.

Country	Region	No. Tested	No. Positive	% Positive	Test	Cut-Off Titer	Remarks	Reference
Algeria	5 districts	350	51	14.6	ELISA ^a1^		Slaughter. Age, housing, presence of other animals AS ^b^.	[25]
Australia	South	2114	209	9.9	MAT ^c^	25	Wild rabbits from 12 sites over a 12-year period.	[20]
Brazil	Minas Gerais	21	2	9.5	MAT	25	Viable Tg ^d^ isolated from 1 rabbit.	[23]
Brazil	Pernambuco	150	10	6.7	MAT	25	Slaughter and breeding farms. Tg DNA.	[26]
Brazil	São Paulo	74	1	1.3	ELISA	10	Meat juice tested in commercial meat cuts.	[16]
China	Hebei	175	22	12.6	IHA ^e1^	64	Titers 1:64 in 20, 1:256 in 2.	[27]
China	Heilongjiang, Inner Mongolia, Jilin, Liaoning	1132	51	4.5	MAT	25	Age AS. Heilongjiang (3.7% of 243), Jilin (4.5% of 354), Liaoning (6.5% of 246), Inner Mongolia (3.5% of 289).	[10]
China	Henan	1213	128	10.5	MAT	25	Age AS.	[4]
China	Shanghai	77	18	23.4	ELISA	100	Viable Tg isolate.	[28]
Czech Republic, Slovakia	6 regions	681	103	5.5	ELISA	20	Farmed. Slaughter. Age, management AS.	[9]
Egypt	Cairo, Qalyubia, Sharkia	150	40	26.7	ELISA^1^		Slaughter animals. IgM antibodies in 9 (6.0%). Age, presence of cats and management AS.	[29]
Egypt	North	64	24	37.5	IHA	80	Titers 1:640 in 8, 1:320 in 8 and 1:80 in 8.	[30]
Egypt	North	194	22	11.3	IHA^2^	80	Farmed rabbits. Coinfection with *Encephalitozoon cuniculi*.	[31]
Italy	Veneto	260	38	14.6	IFAT ^f^	50	Farmed rabbits. Breed, geography AS.	[18]
Japan	20 prefectures	337	3	0.9	LAT ^g1^	32	Pet rabbits. Results confirmed by IFAT, Western blot, and ELISA.	[17]
Korea	Chungnam, Junbuk	142	15	10.6	ELISA		Breeding farms. Sex, age NA ^h^. Tg DNA in 23 (16.2%) in blood.	[32]
Mexico	Durango	429	70	16.3	MAT	25	Prevalence higher in semi-cold climate, raised in backyard and in young (0.3 to 2-month-old rabbits).	[22]
Poland	Lublin	12	2	16.7	MAT^1^	40	Viable Tg isolated from brain of rabbit; the owner ate raw rabbit meat.	[33]
Poland	3 regions	360	44	12.1	MAT^1^	25	Negative PCR in blood. Contact with cats and fed unwashed vegetables AS.	[34]
Portugal	South	36	1	2.8	MAT^1^	20	Wild rabbits	[35]
Scotland	Perthshire	548	18	3.3	MAT	20	Wild rabbits surveyed. Sex, heavy infection with liver coccidiosis AS.	[36]
Spain	Andalusia	85	14	11.9	MAT	25	Wild rabbits sharing habitat with endangered Iberian lynx.	[37]

^a^ ELISA = Enzyme-linked immunosorbent assay. Unless stated otherwise, ELISA = ELISA in-house. ELISA^1^ Rabbit *Toxoplasma* IgM and IgG antibody ELISA kits (MyBioSource, California, USA). ^b^ AS = Association. ^c^ MAT = Modified agglutination test [38]. MAT^1^ (Toxo-Screen DA^®^, Biomérieux, Lyon, France). ^d^ Tg = *Toxoplasma gondii*. ^e^ IHA = Indirect hemagglutination assay. IHA^1^ (Lanzhou Veterinary Research Institute of Chinese Academy of Agricultural Sciences, Lanzhou, China). IHA^2^ (Toxo-IHA Fumouze, Laboratories, Fumoze Division Diagnostics, France). ^f^ IFAT = Indirect fluorescent antibody test. ^g^ LAT = Latex agglutination test. LAT^1^ (Toxocheck-MT, Eiken Chemical, Tokyo, Japan). ^h^ NA = No association.

**Table 2 microorganisms-09-00597-t002:** Prevalence of *Toxoplasma gondii* in hares (*Lepus* spp.) from 2010 to 2020.

Country	Region	No. Tested	No. Positive	% Positive	Test	Cut-Off Titer	Remarks	Reference
Austria	Lower Austria, Salzburg, Tyrol	383	48	13.0	IFAT ^a^	40	European hare (Lepus europaeus). Hunted from 2004–2007. 4.0% N. caninum ^b^ coinfection.	[21]
China	Hubei	331	17	5.1	IHA ^c1^	64	Cape hare (Lepus capensis).	[39]
China	Shandong	358	29	8.1	IHA^1^	64	Tolai hares (Lepus tolai) slaughtered in rural markets. Tg ^d^ DNA and genotype.	[14]
Czech Republic	3 districts	333	71	21.0	IFAT	40	European hare (Lepus europaeus). Hunted from 2004–2007. 8.0% N. caninum coinfection.	[21]
Finland	NS ^e^	107	0	0	MAT ^f1^	40	European hare (Lepus europaeus).	[40]
Finland	NS	96	4	4.2	MAT	40	Mountain hare (Lepus timidus).	[40]
France	NS	23	2	9.0	MAT	25	European hare (Lepus europaeus). Hunted from 2003–2008. Tg not isolated from seropositive hares.	[41]
Greece	Central, North	105	6	5.7	IFAT	40	European hare (Lepus europaeus). Free-ranged. Tg DNA not detected in livers of 52 hunted.	[42]
Italy	Pisa	222	3	1.3	MAT^1^	40	European hare (Lepus europaeus). Free-ranged. 0 of 81 from mountains, 3 of 141 from plains.	[43]
Slovakia	Nitra	209	13	6.0	IFAT	40	European hare (Lepus europaeus). Hunted from 2004–2007.	[21]
Spain	Navarra	298	34	11.4	MAT	25	Iberian hare (Lepus granatensis); Hunted from 2009–2011. Higher prevalence in juveniles. Location and year sampling AS ^g^. No PCR positive in liver.	[5]

^a^ IFAT = Indirect fluorescent antibody test. ^b^
*N. caninum = Neospora caninum.*
^c^ IHA = Indirect hemagglutination assay. IHA^1^ (Lanzhou Veterinary Research Institute, Chinese Academy of Agricultural Sciences, Lanzhou, China). ^d^
*Tg = Toxoplasma gondii*. ^e^ NS = Not stated. ^f^ MAT = Modified agglutination test [38]. MAT^1^ (Toxo-Screen DA^®^, Biomérieux, Lyon, France). ^g^ AS = Association.

**Table 3 microorganisms-09-00597-t003:** Isolation of viable *Toxoplasma gondii* from rabbits.

Country	Region	No.	Antibodies	Bioassay	Isolate Designation	Genotyping	Notes	Reference
Argentina	Buenos Aires	1	IFAT ^a^, 1:2800	Brain in SW ^b^ mice	Rabbit 2Arg	9 PCR-RFLP ^c^ markers—TOXODB genotype #48	Aborted rabbit doe. Disseminated toxoplasmosis	[44,45]
Brazil	Minas Gerais	2	MAT ^d^, 1:100; 1:40	Brain in SW mice	TgRabbitBr1 TgRabbitBr2	TOXODB genotype #19, TOXODB #11	A cat fed the brain of rabbit 1 excreted *T. gondii* oocysts. Both strains were pathogenic to SW mice	[23]
China	Shanghai	1	ELISA ^e^ -positive	Brain, spleen, liver in BALB/c mice	ND ^f^	4 PCR-RFLP markers—TOXODB genotype #2	Pathogenic to BALB/c mice	[28]

^a^ IFAT = Indirect fluorescent antibody test. ^b^ SW **=** Swiss Webster albino mice. ^c^ PCR-RFLP = Restriction fragment length polymorphism. ^d^ MAT = Modified agglutination test [36a]. ^e^ ELISA = Enzyme-linked immunosorbent assay. ^f^ ND = Not done.

**Table 4 microorganisms-09-00597-t004:** *Toxoplasma gondii* DNA in tissues of rabbits and hares.

Country	Location	Host	No. Tested	Tissues ^a^	PCR (Gene)	No. Positive	% Positive	Genotype	Reference
Brazil	Pernambuco	Domestic rabbit	54	B, D, H	(529bp—TOX4,5 primers)	5	9.2	ND ^b^	[26]
Brazil	Rio Grande do Sul, Serafina Corrêa, Southern	Rabbit	1 (case report) 18 examined	S (frozen)	(529 bp), 11 PCR-RFLP ^c^ markers; 15 MS ^d^ markers	1	100 (clinical case)	ToxoDB genotype #8 (type BrIII). MS: 291 allele for typing marker TUB2 in this genotype	[11]
China	Luoyang	Domestic rabbit meat from food markets	96	B, H	nPCR ^e^ (B1), 11 PCR-RFLP markers	2	2.1	ToxoDB genotype #9 (TgRbHN1)	[7]
	Nanyang		248			8	3.2		
	Zhengzhou		126			3	2.4		
China	Shandong	Tolai hare (*Lepus tolai*) slaughtered in rural markets	358	B	nPCR (B1), 11 PCR-RFLP markers	23	6.4	ToxoDB genotype #9 (TgWR1–4)	[14]
Czech Republic	South Moravia	Hares (*L. europaeus*)	36	H, K, L, Lv, S (in one), H, S in the second	RT-PCR ^f^ (529 bp), 15 MS markers	2	5.5	Type II in the two positive samples (archeotypal type II and type II variant)	[19]
Finland	NS ^g^	Hares (*L. europaeus* and *L. timidus*). Fatal toxoplasmosis	14 *L. europaeus*; 4 *L. timidus*	LN, Lv, S	6 MS markers	18	100 (clinical cases)	ToxoDB genotype #1 (type II). The size of the PCR product at the seventh marker (M48) varied (213–229 base pairs)	[40]
Italy	Piedmont	Cotton tailed rabbits, invasive (*Sylvilagus floridanus*)	144	B, Sk	(ITS1)	3 (B of 1, Sk in 2)	2.1	ND	[13]
Korea	Chungnam, Junbuk	Crossbreed, Flemish giant, chinchilla, New Zealand white (pets breeding farms)	142	Bl ^h^	nPCR (B1)	23	16.2	ND	[32]
Portugal	Central	Wild hunted rabbits, *O. cuniculus*	28	D, H	nPCR (SAG2)	19	67.9	ND	[46]

^a^ B = Brain, Bl = Blood, D = Diaphragm, H = Heart, K = Kidney, L = Lung, LN = Lymph nodes, Lv = Liver, S = Spleen, Sk = Skeletal muscle. ^b^ ND = Not done. ^c^ PCR-RFLP = Restriction fragment length polymorphism. ^d^ MS = Microsatellite. ^e^ nPCR = Nested PCR. ^f^ RT-PCR = Real-time PCR. ^g^ NS = Not stated. ^h^ Whole blood from live rabbits.

**Table 5 microorganisms-09-00597-t005:** Summary of reports of clinical toxoplasmosis in rabbits and hares from 2010 to 2020.

Species	Source	Remarks	Histology	IHC ^a^	Reference
Brown hare (*Lepus europaeus)*	14 retrospective study, Finland	Retrospective study of wild brown hares that died. Genotyped (see Table 4).	Yes	Yes	[40]
	2 found dead, Czech Republic	Acute toxoplasmosis, one animal with more than 1 × 10^6^ parasites/g of tissue. Genotyped (see Table 4).	Yes	Yes	[19]
Mountain hare *(Lepus timidus)*	4 retrospective study, Finland	Retrospective study of mountain hares that died. Genotyped (see Table 4).	Yes	Yes	[40]
Rabbit (*Oryctolagus cuniculus*)	1 domestic, Brazil	Fatal acute disease. Genotyped (see Table 4).	Yes	Yes	[11]
	1 aborted doe on a farm, Argentina	Viable isolation. Genotyped [43] (see Table 4)	Yes	Yes	[44]

^a^ IHC = Immunohistochemistry.

## Data Availability

Data sharing not applicable. No new data were created or analyzed in this study.

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
