# Peer review of "Epidemiological and Public Health Significance of Toxoplasma gondii Infection in Wild Rabbits and Hares: 2010–2020"

_microorganisms, 2021, doi:10.3390/microorganisms9030597_

Round 1

Reviewer 1 Report

In this manuscript the authors revise the epidemiological and public health significance of Toxoplasma gondii infection in wild rabbits and hares between 2010-2020. In my opinion, this is an interesting work that will provide updated insights on this subject.

I have some minor comments:

1. Please write Toxoplasma gondii and T. gondii in italics. Confirm throughout the manuscript.

2. When indicating the cut-off titer, please use the inverse of the dilution. A titer should not be indicated as a dilution. Please correct throughout the manuscript.

3. Scientific names should be in italics. e.g.point 2.2. L. europaeus, L. timidus. This should be observed throughout the entire manuscript.

Author Response

Answer to Reviewer 1. this manuscript the authors revise the epidemiological and public health significance of Toxoplasma gondii infection in wild rabbits and hares between 2010-2020. In my opinion, this is an interesting work that will provide updated insights on this subject.

I have some minor comments:

  1. Please write Toxoplasma gondii and T. gondii in italics. Confirm throughout the manuscript. Done. Somehow the italics went wrong when the manuscript was submitted to the journal.
  2. When indicating the cut-off titer, please use the inverse of the dilution. A titer should not be indicated as a dilution. Please correct throughout the manuscript. Indicated the cut-off as the inverse of the dilution in Table 1 and 2.
  3. Scientific names should be in italics. e.g.point 2.2. L. europaeus, L. timidus. This should be observed throughout the entire manuscript. Done. Somehow the italics went wrong when the manuscript was submitted to the journal

Reviewer 2 Report

This is an interesting and well-written review. The information on the seroprevalence, clinical cases, diagnosis perfomance and genotypes reports, provide an usefull up to date.

In my opinión the review is appropriate to be published with minor comments; concretely, these are some of the punctual specifications or suggestions:

-T. gondii, Toxoplasma gondii, and other scientific names (ie Lepus spp in the heading of Table 2) should be in italics. Please verify this in the text of the manuscript.

--What does "1 rabbit" mean in table 2_column "host"? Maybe the number "1" shouldn't be there.

Author Response

Answer to Reviewer 2.

This is an interesting and well-written review. The information on the seroprevalence, clinical cases, diagnosis performance and genotypes reports, provide an usefull up to date.

In my opinion the review is appropriate to be published with minor comments; concretely, these are some of the punctual specifications or suggestions:

-T. gondii, Toxoplasma gondii, and other scientific names (ie Lepus spp in the heading of Table 2) should be in italics. Please verify this in the text of the manuscript. Done. Somehow the italics went wrong when the manuscript was submitted to the journal

--What does "1 rabbit" mean in table 2_column "host"? Maybe the number "1" shouldn't be there. Eliminated # 1

Reviewer 3 Report

The review of the presence of T. gondii in rabbits and hares shows us that this microorganism is still present with a greater or lesser proportion in different geographical regions.

In the work, studies published in the last 10 years have been collected, as well as the analysis technique used to evaluate the presence of the microorganism.

Although the review has made clear the prevalence of this microorganism in rabbits and hares, I believe that it can be improved in some section.

In the introduction section, as it is a review, the authors could reflect the mechanism of action of T. gondii, as well as the transmission cascade involved in its infection. Since later in the PCR detection section they refer to some involved receptors.

Likewise, in this introduction section, the infection mechanisms through which this microorganism can distribute itself could be briefly considered.

Table 1 shows the analysis methodologies and also in section 5. The authors could compare or reflect on the sensitivity of each of them in order to obtain one or more conclusions about the presence of the microorganism.

On page 6, section 2.2 and 2.1, the authors forgot to italicize the scientific names (L. europeus, T. gondii, L. Timidus, L. granatensis)

In section 3 the genotypes are reflected but no information on the difference between them is included

In section 4, similar to the previous comment, the authors could reflect where the amplified part is located

In the conclusions section, a hypothesis of those factors that favor the disease to a greater or lesser extent could be reflected

The review is interesting because in addition to including the analysis techniques for the identification of the microorganism, the distribution in different countries is reflected, it also directs us to studies where the reader can go to perform their analyzes.

Author Response

Answer to Reviewer 3

The review of the presence of T. gondii in rabbits and hares shows us that this microorganism is still present with a greater or lesser proportion in different geographical regions.

In the work, studies published in the last 10 years have been collected, as well as the analysis technique used to evaluate the presence of the microorganism.

Although the review has made clear the prevalence of this microorganism in rabbits and hares, I believe that it can be improved in some section.

In the introduction section, as it is a review, the authors could reflect the mechanism of action of T. gondii, as well as the transmission cascade involved in its infection. Since later in the PCR detection section they refer to some involved receptors. Likewise, in this introduction section, the infection mechanisms through which this microorganism can distribute itself could be briefly considered. A paragraph has been included in the introduction section about the mechanisms of transmission, parasitic stages and immune response of this infection, as well as the tissue distribution of the parasite (marked in red in the introduction section).

Table 1 shows the analysis methodologies and also in section 5. The authors could compare or reflect on the sensitivity of each of them in order to obtain one or more conclusions about the presence of the microorganism. A paragraph has been included after table 1 (marked in red) indicating the use of the characteristics of the different serological methods used for studies in rabbits and hares, and the fact that not many methods have been validated in many studies in most animal species in general, as well as in rabbits and hares, and that development and standardization of diagnostic tests for T. gondii infection is still necessary.

On page 6, section 2.2 and 2.1, the authors forgot to italicize the scientific names (L. europeus, T. gondii, L. Timidus, L. granatensis). Done. Somehow the italics went wrong when the manuscript was submitted to the journal

In section 3 the genotypes are reflected but no information on the difference between them is included. A sentence was added on the fact that little is still known of strains and genotypes in rabbits and hares (marked in red).

In section 4, similar to the previous comment, the authors could reflect where the amplified part is located. Added that “The most frequently used target genes for T. gondii PCR in both humans and animals are the repetitive regions of the 35-copy B1 gene and 300-copy 529 bp repetitive element” (marked in red).  

In the conclusions section, a hypothesis of those factors that favor the disease to a greater or lesser extent could be reflected. Added that more studies of the specific factors that exacerbate disease in rabbits and hares are also needed (marked in red).

The review is interesting because in addition to including the analysis techniques for the identification of the microorganism, the distribution in different countries is reflected, it also directs us to studies where the reader can go to perform their analyzes.

Thank you for your nice comments.